# Effectiveness and acceptability of cognitive–behavioural therapy based interventions for maternal peripartum depression: a systematic review, meta-analysis and thematic synthesis protocol

Danelle Pettman [1], Heather O'Mahen [2], Agneta Skoog Svanberg [3], Louise von Essen [1], Cathrine Axfors [3], Oscar Blomberg,[1] Joanne Woodford [1]

¹Clinical Psychology in Healthcare, Department of Women's and Children's Health, Uppsala University, Uppsala, Sweden
²Mood Disorders Centre, School of Psychology, University of Exeter, Exeter, UK
³Reproductive Health, Department of Women's and Children's Health, Uppsala University, Uppsala, Sweden

**Correspondence to**
Dr Danelle Pettman;
danelle.pettman@kbh.uu.se

## ABSTRACT

**Introduction** Peripartum depression is a common mental health difficulty associated with a range of negative impacts for the mother, infant and wider family. This review will examine the effectiveness of cognitive–behavioural therapy (CBT) based interventions for peripartum depression. Secondary aims are to explore the effect of CBT-based interventions targeted at peripartum depression on novel secondary outcomes and moderators potentially associated with effectiveness. To date, there has been little examination of effect on important secondary outcomes (eg, anxiety, stress and parenting), nor clinical and methodological moderators. Further, this review aims to explore the acceptability of CBT-based interventions for women with peripartum depression and examine important adaptations for this population.

**Methods and analysis** Electronic databases (e.g., MEDLINE; ISI Web of Science; CINAHL; CENTRAL; Prospero; EMBASE; ASSIA; PsychINFO; SCOPUS; And Swemed+) will be systematically searched. Database searches will be supplemented by expert contact, reference and citation checking, and grey literature. Primary outcomes of interest will be validated measures of symptoms of depression. A proposed meta-analysis will examine: (1) the overall effectiveness of psychological interventions in improving symptoms of depression (both self-reported and diagnosed major depression) in the peripartum period; (2) the impact of interventions on secondary outcomes (eg, anxiety, stress and parenting); (3) clinical and methodological moderators associated with effectiveness. A thematic synthesis will be conducted on qualitative data exploring the acceptability of CBT-based intervention for postpartum depression including participants' experience and perspectives of the interventions, satisfaction, barriers and facilitators to intervention use, intervention relevance to mothers' situations and suggestions for improvements to tailor interventions to the peripartum client group.

**Ethics and dissemination** Formal ethical approval is not required by the National Ethical Review Board in Sweden as primary data will not be collected. The results will be disseminated through a peer-reviewed publication and inform the development of a new psychological intervention for peripartum depression. This study including protocol development will run from March 2019 to March 2020.

## Strengths and limitations of this study

► Systematic review adopts quality standards informed by the Centre of Reviews and Dissemination guidance and Preferred Reporting Items for Systematic Reviews and Meta-Analyses Protocols guidelines, such as independent study selection, data extraction and risk of bias assessments by two researchers.

► The review examines the effectiveness of cognitive–behavioural therapy (CBT) based peripartum depression interventions on secondary outcomes (eg, anxiety, stress and parenting) as well as clinical and methodological moderators of effectiveness of CBT-based interventions for peripartum depression.

► The review is the first to synthesise qualitative data regarding participants experience and perspectives of CBT-based interventions for peripartum depression.

► High levels of clinical heterogeneity may exist as a consequence of included studies defining and delivering CBT-based interventions in a variety of ways.

► Due to resource limitations, selected studies are only those publicly available in the English or Swedish language; therefore, language bias may be present.

## INTRODUCTION

Peripartum depression (PPD) is a common mental health difficulty characterised by low mood during the pregnancy (antepartum period) and/or after childbirth (postpartum period).[1] While it has been argued depression occurring during pregnancy is distinctly different to postpartum depression[2] in terms of epidemiological,[3] hormonal sensitivity[4] and immune system functioning[5] for the purposes

of this review, we will use the more inclusive 'peripartum' definition and examine time point of intervention (eg, antepartum or postpartum period) as a potential moderator of effect.

Prevalence rates of PPD vary dependent on whether measured via screening scales or standardised diagnostic tools,[6] with the former at a rate of 19.2% and the latter with at a rate of 14.5%.[7] A recent study indicates that symptoms of depression are more prevalent during the postpartum period than in age-matched controls.[8] Further, approximately 40% of women will experience their first depressive episode postpartum,[9] and when left untreated, are more likely to experience further episodes of depression later in life.[10] The impact of PPD is significant for both mothers, infants and the wider family. For mothers, PPD is associated with poor health-related quality of life and in extreme cases loss of life through suicide.[11] Further, postpartum depression is associated with increased risk for impairments in the mother's perceived bonding and attachment to the infant, both during the antepartum and postpartum period; in the postpartum period, this is related to impairments in maternal responsiveness or sensitivity (hereafter referred to as parenting).[12] Prospectively, PPD is related to impairments in the infant's cognitive, social and emotional functioning.[12] [13] Maternal postpartum depression is also associated with increased marital discord[14] and depression in fathers.[15] Given the prevalence and impact of PPD, it is important to identify and develop effective interventions.

Existing reviews highlight the effectiveness of a number of interventions for PPD including medication (eg, antidepressant medication), health promotion interventions (eg, educational home visits) and psychological interventions (eg, cognitive–behavioural therapy (CBT) and interpersonal therapy.[16–27] However, the cost:benefit ratio for medication shifts for PPD given unclear evidence concerning safety during the pregnancy,[28] and mothers' concerns regarding medication use while breast feeding.[29] Further, women experiencing PPD indicate a preference for psychological interventions over medication.[29] Although there are mixed findings regarding the superiority of one type of psychological intervention over another,[17 25 27] several reviews have found CBT to be an effective psychological intervention for PPD.[23 25 26 30] As such CBT-based interventions are recommended as a first-line intervention in clinical guidelines for a number of common mental health difficulties.[31] Additionally, in comparison with other therapeutic approaches, CBT is provided in a wider range of intervention formats including; individual,[32] group[33] and internet administered.[34]

While existing interventions have demonstrated CBT-based interventions for PPD to be effective for depression they have often neglected important secondary outcomes, for example, anxiety, stress (individual and perceived parenting), parenting (eg, sensitivity/responsiveness) perceived social support and perceived parental competence has been largely unexamined. It is important to examine the effect of interventions for PPD on symptoms of anxiety given comorbidity rates[35] and research suggesting both maternal anxiety and depression should be addressed in interventions.[36] An examination of effect on stress (individual and perceived parenting) is also warranted given the deleterious impact of stress on mothers and infants.[37] Indeed, perceived parenting stress disrupts the ability of mothers to appropriately assess infant signals and to react sensitively to them.[38] Further, research suggests interventions for PPD have little benefit on child outcomes,[39] and therefore, both maternal depression and parenting difficulties should be addressed.[40] Additionally, perceived low levels of parental competence[41] and poor social support are associated with PPD suggesting interventions should aim to improve both and be included as a secondary outcome.[42]

A further omission in the current evidence base of interventions for PPD concerns clinical and methodological moderators potentially associated with effectiveness. While a recent review has explored the effect of a variety of moderators on CBT for PPD, including type of study control condition (eg, wait-list control or treatment-as-usual (TAU)) and time point of intervention (eg, antepartum or post partum)[26] a number of potentially important moderators were not examined. As such, the present review will facilitate the exploration of a number of novel moderators, for example, the potential effect of health professional delivering intervention (eg, psychologists, family doctors, nurses and non-professionals). Investigating the potential impact of these moderators is of importance given CBT-based interventions are delivered by a wide range of health professionals.[30] In addition, we will examine the potential effect of including interventions that also include components targeting parenting. This is of particular importance given that research has consistently associated parenting difficulties with postnatal depression.[43] However, to date, to the best of our knowledge, existing reviews have not examined the inclusion of parenting components within CBT-based interventions as a moderator. Increased understanding of clinical moderators associated with effectiveness are particularly important to inform the development of future interventions for the population.

In addition to the analysis of secondary outcomes and clinical and methodological moderators of the effect of intervention for PPD on the symptoms of depression, further research is also needed to examine intervention acceptability and how barriers to accessing psychological interventions during the peripartum period are addressed. A recent study found low rates of intervention engagement in PPD populations in clinical settings, with up to 86% of those suitable for psychological interventions failing to receive any support.[44] Previous qualitative studies have highlighted both practice (eg, transportations difficulties, attending appointments around feeding and sleep schedules, childcare needs) and psychological (stigma, sleep deprivation, fear of having child removed) barriers.[23 45] Although there is some evidence suggesting

interventions adapted to address these barriers, can improve intervention engagement[19] there is a lack of systematic research investigating how such barriers are addressed by interventions and how uniquely tailoring interventions to the peripartum period may increase engagement and acceptability. As such, we will examine qualitative studies preceding eligible randomised controlled trails (RCTs) regarding intervention development and qualitative studies embedded within RCTs[46] to describe participants' experiences, perspectives and adaptations made to tailor CBT interventions to mothers'.

This review seeks to provide a systematic review of CBT-based interventions for PPD. Meta-analysis, data permitting, will be used to examine the effect of CBT on the main outcomes (symptoms of depression and depression diagnosis) and secondary outcomes (eg, anxiety, stress and parenting). In addition, this review will explore the potential clinical and methodological moderators on the symptoms of depression. Further, data from preceding or embedded qualitative studies[46] will be synthesised to examine women's experiences and perspectives of the interventions. Together this will provide a comprehensive review of the best current evidence regarding CBT-based interventions for PPD that can be used to inform future intervention development.

### Objectives

(1) to examine the effectiveness of CBT-based interventions for PPD on symptoms of depression; (2) to examine the effectiveness of CBT-based interventions for PPD on anxiety, stress (individual and perceived parenting stress), parenting (eg, sensitivity/responsiveness) perceived social support and perceived parental competence; (3) to investigate clinical and methodological moderators potentially associated with effectiveness; and (4) to describe the acceptability of CBT-based interventions for PPD including (but not limited to) participants' experience and perspectives of the interventions, satisfaction, barriers and facilitators to intervention use, intervention relevance to mothers' situations and suggestions for improvements and (5) to identify adaptations to tailor CBT-based interventions to mothers' situations potentially associated with acceptability.

## METHODS AND ANALYSIS
### Design

The protocol for this systematic review has been developed in accordance with the Preferred Reporting Items for Systematic Reviews and Meta-Analyses Protocols (PRISMA) 2015 statement[47] (see online supplementary appendix 1) This systematic review will be conducted in accordance with the Cochrane Collaboration guidance for systematic reviews[48] and reported in accordance with the PRISMA statement.[49]

### Patient and public involvement

Due to project funding constraints, patients and members of the public were not involved in the design of this protocol. However, results of the present review will be used to inform the further development of a PPD intervention, following the Medical Research Council (MRC) complex interventions framework[50] targeting both depression and parenting. We will adopt a participatory action research approach[51] to inform the development of the intervention, placing mothers with lived experience of PPD at the centre of the intervention development process.

### Inclusion and exclusion criteria

Study inclusion and exclusion criteria were developed using the following domains: Patient, Intervention, Comparison, Outcome and Study Design (PICOS).[52]

### Participants

Adult women aged ≥16 years (a cut-off of 16 years was chosen as studies are likely to be drawn from international settings with variation in the age at which someone is deemed an adult).[53 54] From pregnancy to 12 months post partum with either; (1) a diagnosis of PPD, for example, Diagnostic and Statistical Manual of Mental Disorders IV or V[1 55]; or (2) reporting depression symptomatology within the peripartum period (from pregnancy to 1-year post partum) using a validated tool, for example, Edinburgh Postnatal Depression Scale (EPDS)[56]. No limits will be placed on the severity of depression at baseline. Exclusion criteria are: (1) intervention for mood disorders other than depression (eg, bipolar affective disorder) and (2) intervention explicitly targeting the prevention of symptoms of depression in at-risk mothers in the perinatal period.[23] For example, a study with the aim to treat current depression during the antepartum period, with an aim to prevent depression during the postpartum period would be eligible for inclusion. However, interventions explicitly targeting the prevention of depression during either the antepartum or postpartum period will be excluded.

### Interventions

Only studies investigating interventions explicitly targeting PPD will be included. Eligible interventions will state the use of CBT-based interventions, including behavioural activation or problem solving, and contain specific therapeutic components associated with these approaches. These approaches have been previously synthesised in a meta-analysis for psychotherapy in major depression.[17] CBT will be defined as interventions in which the focus is modifying a client's dysfunctional thoughts on current behaviour and future functioning.[17] Behavioural Activation (BA) will be defined as interventions targeting reductions in behavioural avoidance and increases in positively reinforcing activities, including interventions that focus on scheduling behaviours.[57] Problem-solving interventions will be defined as a psychological

intervention including the following elements: definition of personal problems, generation of multiple solutions to each problem, selection of the best solution, developing a systematic plan for this solution and evaluating whether the solution has resolved the problem.[17] There will be no exclusions regarding professional group supporting the intervention, the clinical setting of the intervention, the method of delivery (individual, group or internet administered), or support methods used (eg, email, face to face, telephone). Additionally, self-guided/self-administered interventions will be eligible for inclusion.

## Comparators

Control conditions eligible for inclusion will be based on standard definitions,[58] with eligible control conditions including: (1) no-treatment control; (2) wait-list control; (3) TAU; (4) non-specific factors component control; (5) specific factors component control and (6) active comparator. Only trial designs that allow for the isolation of the effects of CBT will be included given it is important for active comparators to discriminate intervention effects.[59] For example, a study comparing CBT alone versus medication alone would be excluded as it would not be possible to isolate the effect of the CBT. However, CBT plus medication versus medication alone and CBT plus information versus information alone would be eligible for inclusion.

## Outcomes

This is a mixed-methods systematic review involving both quantitative and qualitative outcomes.[60]

### Quantitative outcomes

Studies eligible for inclusion will use a self-report or proxy/clinician administered standardised measurement of depression (eg, the Beck Depression Inventory)[61] or PPD (eg, EPDS).[56] In order to ensure the quality of the outcome measurements concerning the primary outcome of depression only studies using depression outcome measurements with at least acceptable internal consistency and test–retest reliability at Cronbach's alpha ≥0.70 (as reported in outcome measurement validation studies) will be included.[62] In addition the proportion of mothers meeting diagnostic criteria for depression following intervention will be assessed. In order to ensure quality, only diagnosis made with either a stuctured clinical interview for diagnositic and statistical manual (SCID)[63] or Mini-International Neuropsychiatric Interview[64] will be included, as a recent systematic review has highlighted that only these two tools meet sensitivity and specificity criteria.[65] Secondary outcomes of interest are validated self-report measurements of (1) anxiety, (eg, the Beck Anxiety Inventory);[66] (2) individual stress (eg, Perceived Stress Scale);[67] (3) perceived parental stress (eg, Parenting Stress Index);[68] (4) self-report parenting (eg, sensitivity/responsiveness such as; Postpartum Bonding Questionnaire);[69] 5) perceived social support (eg, Multidimensional Scale of Perceived Social Support)[70]; and

(6) parental competence (eg, Parenting Sense of Competence Scale).[71] Additionally, observational parenting (eg, sensitivity/responsiveness) measures (eg, video tapes assessed with mind-mindedness coding manual)[72] will be included as secondary outcome measurements.

### Qualitative outcomes

Studies eligible for inclusion will be: (1) qualitative studies preceding RCT studies to develop the intervention(s) examined in RCTs or qualitative studies embedded within RCT studies and (2) examining acceptability of interventions (eg, participants' experience and perspectives of the interventions, satisfaction, barriers and facilitators to intervention use, intervention relevance to mothers' situations and suggestions for improvements); and/or (3) examining adaptations made to tailor the intervention to the population. If mixed-methods studies are identified using quantitative measures of intervention acceptability (eg, the Treatment Acceptability/Adherence Scale[73]), these data will be extracted and triangulated with the qualitative findings regarding acceptability.

## Study designs

Both studies with quantitative and qualitative designs will be included.

### Quantitative

To examine effectiveness, only RCTs will be included, with non-randomised and uncontrolled designs excluded. RCT designs were chosen for this systematic review as they are less prone to biases inherent with other study designs.[74] Randomisation aims to balance known and unknown variables between the treatment groups in order to control for confounding. Random allocation also facilitates the blinding of interventions, for example, to evaluators.[75] Meta-epidemiological studies have shown that trials without proper RCT design, that is, with inadequate concealment of treatment allocation or inadequate randomisation, tend to report higher effect sizes.[76]

Further, RCTs with randomisation procedures at allocation and concealment rated as high risk of bias, in line with the revised Cochrane risk-of-bias tool for randomised trials (RoB 2),[77] will be excluded (see online supplementary appendix 2). This is to minimise the inclusion of studies of low quality with high risk of selection bias known to inflate effect sizes[78 79] and is a technique used in previous systematic reviews and meta-analyses.[80–82]

### Qualitative

Data from preceding or embedded qualitative studies within included RCTs will be examined for mothers' experiences and perspectives of interventions and potential adaptations of interventions.

## Search methods
### Electronic searches

Searches will be carried out in relevant international electronic databases; MEDLINE; ISI Web of Science; Cumulative Index to Nursing and Allied Health Literature;

Cochrane Central Register of Controlled Trials; Prospero; Excerpta Medica DataBase (EMBASE); Applied Social Sciences Index and Abstracts; PsycINFO; SCOPUS and SweMed+. Searches will also be conducted on clinical trial registers (www.ClinicalTrials.gov and www.who.int/trialsearch/) and conference proceedings (BIOSIS Previews; Index to Scientific and Technical Proceedings, Health Management Consortium and Web of Science with Conference Proceedings). Databases will be searched using medical subject headings and text words in the title and abstract (see online supplementary appendix 3). The search strategy was developed alongside Agnes Kotka (librarian at Uppsala University Library) and reviewed by Professor Alkistis Skalkidou (Uppsala University) using the PRESS Peer Review Guidelines[83]; see online supplementary appendix 4). No date limits will be imposed. Studies published in English and Swedish will be accepted.

### Other resources

Forward citation searches using Google Scholar (forward citation chasing)[84] will be conducted for all included studies and reference lists of all of the included studies will be hand searched. Studies included within identified secondary evidence reports (eg, relevant systematic reviews and meta-analyses) will also be reviewed. Systematic reviews will only be included in the hand search if they meet the following criteria: (1) search of at least one database; (2) reporting of selection criteria; (3) a quality appraisal of included studies and (4) provide a list and synthesis of included studies.[85]

Additionally, the five journals containing the highest numbers of included studies will be hand searched for recent potentially eligible publications in the past 12 months.[86] Grey literature will be searched using; OpenGrey, a system for grey literature produced in Europe, such as research reports, doctoral dissertations and conference papers (http://www.opengrey.eu/); ProQuest a dissertation and thesis database (http://www.proquest.com/en-US/catalogs/databases/detail/ pqdt.shtml); and Digitala Vetenskapliga Arkivet DIVA a portal for searching publications from Swedish universities managed by Uppsala University. Searches for relevant dissertations will be conducted; however, due to time and funding limitations full dissertations will not be reviewed. The references for potentially eligible dissertations will be made available as part of a list of excluded studies. Potentially relevant studies published in a languages other than English and Swedish will not be included within the review due to lack of resources for translation, and will be provided as part of a list of excluded studies.

### Screening and data extraction
#### Screening

Duplicates of studies across searches will be identified and removed. Disagreements regarding inclusion will be discussed between the two reviewers, with a third reviewer consulted if consensus cannot be reached. Two independent reviewers will conduct a broad screen of study titles and abstracts, followed by full paper checks of potentially eligible studies. Studies will be excluded that do not clearly meet the outlined PICOS criteria. Overall reasons for exclusion will be recorded (see online supplementary appendix 2) and reported on the PRISMA flow chart in the results manuscript. In addition, a more detailed exclusion table will be presented in the result manuscript with inclusion/exclusion presented for each specific criteria in line with the PICOS statement.

In the case of studies in which missing data are needed to determine eligibility, authors will be contacted by email for additional information. For authors who do not respond to the first email within 2 weeks, a follow-up email will be sent.

### Data extraction
#### *Quantitative extraction*

Two reviewers will independently extract data from the included studies following Centre for Reviews and Dissemination guidance.[87] Data will be extracted according to data extraction form (see online supplementary appendix 5), including (1) study characteristics; (2) participant characteristics; (3) intervention characteristics; (4) outcome measurements and (5) quantitative results. The following data will be extracted:

1. Study characteristics: citation, publication type (published or unpublished), country of origin, funding source, language, aims and objectives, design and inclusion/exclusion criteria.
2. Participant characteristics: method of depression assessment (eg, structured clinical interview or screening tool), major depression diagnosis (yes/no), recruitment setting (clinical or community), age of mother (mean, SD), age of infant (mean, SD), time point of intervention (eg, antepartum or post partum), baseline anxiety, comorbidities, ethnicity of mother (n, %), relationship status (n, %), educational status (n, %), employment status (n, %), mothers first child (n, %), average household income (n, %), breast feeding (n, %), level of depression severity at baseline (if clinical cut-offs available), depression chronicity years (mean, SD), number of participants invited, number of participants screened, number of participants eligible, number of participants randomised and reasons for non-eligibility.
3. Intervention characteristics: type of CBT intervention (CBT, BA or problem solving), inclusion of social components (eg, peer-support group or involvement of partner; yes/no), inclusion of parenting intervention components (eg, video interaction guidance; yes/no), treatment manual (yes/no), measurement of treatment adherence (yes/no), method of delivery (eg, face to face, group or internet administered), treatment setting (eg, clinic or community), health professional delivering intervention (eg, clinical psychologist or midwife), study-specific training (yes/no), duration of treatment (weeks), number of sessions, length of sessions (minutes), maximum length of treatment

sessions over treatment course (minutes), group size (mean, SD) and type of control condition used (eg, no-treatment control, wait-list control, TAU, non-specific factors component control, specific factors component control and active comparator).

4. Outcome measurements: primary outcome: outcome measure used, participant n, mean, SD and/or SE for each outcome time point collected and measure of outcome quality (Cronbach alpha for internal consistency and test–retest reliability from original validation papers). Secondary outcome(s) (including: anxiety, individual stress, perceived parental stress, self-report parenting, perceived social support, parental competence and observational parenting): outcome measure used, participant n, mean, SD and/or SE for each outcome time point collected.

5. Statistical techniques: power calculation, intention to treat (yes/no), method of dealing with missing data, baseline comparability and for cluster trials, estimates of intracluster correlation coefficients will be gathered).

6. Participant flow: randomised to intervention, randomised to control, lost to follow-up (at each time point measured), analysed intervention (at each time point measured) and analysed control (at each time point measured) and attrition rate.

7. Research ethics: data relating to ethics (eg, ethical approval, ethical issues highlighted).

### *Qualitative extraction*

Qualitative studies (preceding or embedded within included RCT studies) focused on CBT for PPD intervention acceptability and intervention adaptations will be sought by two reviewers. Data from these studies will be imported into NVivo[88] software for analysis.

Discrepancies will be discussed between the two reviewers, with a third reviewer consulted if consensus cannot be reached. Study authors will be contacted in the event of missing data.

### Risk of bias assessment

Methodological quality of included studies will be assessed using the Cochrane Collaboration's Risk of Bias tool 2.0.[77] Two reviewers will work independently to determine the adequacy of (1) randomisation; (2) allocation to intervention; (3) adherence to intervention; (4) handling of missing outcome data; (5) measurement of outcome and (6) selection of the reported results. Ratings will be compared and any discrepancies discussed, and if consensus is not reached, further discussion will be held with a third reviewer. To assess reporting bias study protocols will be sought for all included studies either via published protocols, trial databases or emailing study authors. Comparisons will be made between outcome measurements reported in the protocol and the paper as well as outcomes reported in the methods and results section of published trial results. Study authors will be contacted to clarify discrepancies and to identify potential

changes to the study protocol and request any missing data. Consistent with a previous meta-analysis,[81] in order to include quality as a moderator each area addressing risk of bias will be given a score of 0, 1 or 2, indicating bias based on the Risk of Bias tool 2.0,[77] yielding a possible total score of 12. The ratings will be defined as high (9–12), unclear (4–8) or low risk (0–3), respectively. All findings will be summarised within a table to allow easy comparison across studies.

### Data synthesis and statistical analysis

Quantitative and qualitative data will be analysed separately.[89] Quantitative data will be managed using an extraction database developed in Microsoft Access for this review and analysed in Comprehensive Meta-analysis (V.2). Qualitative data will be managed and analysed using NVivo V.10.[88]

### Measures of intervention effect

If data allow, a meta-analysis will be conducted. Hedges' g will be calculated to determine the post-treatment between-group standardised mean effect size from outcomes relating to the primary (depression) and secondary outcomes (anxiety, individual stress, perceived parental stress, parenting, perceived social support and parental competence).

Where multiple time points are reported, a primary time point of ≤6 months post-treatment will be adopted for the primary analysis to reduce the likelihood of bias associated with examining short-term post-treatment effects only that are likely to result in elevated effect sizes.[80 90 91] However, length of follow-up will be included as a moderator. The control condition sample size will be halved for studies where two interventions eligible for inclusion are compared with one control condition and comparisons will be analysed separately. Additionally, in studies comparing two control conditions with one intervention condition, comparisons will be analysed separately with the sample size in the intervention condition halved.[92]

### Dealing with missing data

Study authors will be contacted to provide missing means and SD of post-treatment measurement scores. Intention-to-treat principles stipulate the inclusion of every subject who is randomised according to randomised intervention assignment[93] and will be followed as far as possible. In instances in which intention-to-treat data are not available completer data will be extracted.

### Assessment of heterogeneity

Heterogeneity is anticipated due to variations in interventions in the studies, participant characteristics and methodological factors, therefore, a random-effects model[94] will be adopted with Q and $I^2$[95] reported alongside CIs as measures of heterogeneity. If this assumption is not met in the Q and $I^2$ analysis, a fixed-effects model will be used.

## Sensitivity analysis

A sensitivity analysis regarding the overall effect size of the primary outcome (depression) will be conducted by removal of (1) each study individually from the overall analysis and the effect size recalculated in order to estimate the statistical validity of the effect size;[96] (2) small studies (n≤20 across conditions) to explore the suggestion that small trials tend to report larger treatment benefits than larger trials[97]; and/or (3) studies with high attrition rates (≥30% in at least one arm), given attrition rates of ≥30% are associated with imbalances at baseline and are therefore vulnerable to bias such as clinical and psychosocial differences between groups due to differences between those that leave or remain in the study, which is likely to have an impact on the estimated effect of the exposure.[98]

## Funnel asymmetry

If there is a minimum of 10 studies in any meta-analysis, funnel plot asymmetry will be examined for sources of possible bias (eg, publication bias, language bias, inclusion of small studies with poor methodological quality and heterogeneity).[99 100] Comprehensive Meta-analysis (V.2) will be used to assess possible bias. An estimated effect size taking these biases into account will be calculated using the trim and fill procedure,[101] with an estimated effect size calculated separately for each primary and secondary outcome.

## Moderator analysis

If sufficient data are available, moderator analysis will be undertaken to examine clinical and methodological components and participant characteristics of studies potentially associated with effectiveness. The following moderators will be examined:

### *Methodological moderators*
▶ Risk of bias (eg, high, unclear or low).
▶ Type of control condition used (eg, no-treatment control, wait-list control, TAU, non-specific factors component control, specific factors component control and active comparator).[102]
▶ Length of follow-up (eg, short; post-treatment≤3 months, medium; 3–6 months, long; 7–11 months; extended; 12 months+).

### *Clinical moderators*
▶ Severity of depression at baseline (eg, severe, moderate or mild).
▶ Type of CBT intervention (eg, CBT, BA or problem solving).
▶ Interventions including additional social components (eg, peer-support group or involvement of partner). Social components will be defined as interventions facilitating contact with peers or partners in a way that encourages nondirective (without specific psychological techniques) support, such as helping people to express their experiences and emotions and offering empathy.

▶ Interventions including parenting intervention components (eg, video interaction guidance).[103] Parenting intervention components will be defined as interventions that include specific support in relation to the parent infant relationship this could be via specific sessions with a therapist or via self-help materials.
▶ Method of delivery (eg, individual, group or internet administered).
▶ Time point of intervention (eg, antepartum or postpartum period).
▶ Health professional delivering intervention (eg, clinical psychologist or midwife).

With heterogeneity being anticipated, random effects will be adopted with Q and I² reported as measures of heterogeneity.

## Qualitative data synthesis

Thematic synthesis will be used to analyse qualitative data.[104] In this review, eligible qualitative data will be defined as all text labelled as 'results' or 'findings' in study reports.[104] Similar to analysis of primary qualitative datasets, thematic synthesis involves the systematic coding of data and generating of descriptive and analytical themes. These data will be analysed following a three-step process. First, study data will be coded line by line by the first reviewer and checked by a second reviewer with a coding frame developed from codes derived from the data. Second, similar codes will be grouped together to form new 'descriptive themes'. Disparities or discrepancies in coding and theme development will be resolved through discussion or in consultation with a third reviewer if necessary; with the coding frame adjusted accordingly. Once coding is complete the two reviewers will consider each theme and subtheme and examine the text within them for consistency, code interrelations, and identify any potential conceptual hierarchies. Third, analytical themes will be developed to 'go beyond' the primary reported data by synthesising findings across studies and interpreting their meaning in relation to review objectives. Narrative descriptions of each theme will be provided, alongside supporting quotations. All changes to codes, conceptual realignments, discussions and decisions will be documented as part of the review audit trail. Research supervisors (HAO and JW) have extensive experience in the analysis of qualitative data using a variety of approaches, are fully conversant in the use of NVivo and will provide supervision to the PhD Student (DP) and Research Assistant (OB) conducting the meta-synthesis.

## SUMMARY

This review represents the first review of interventions for PPD that includes mixed methods to examine the overall effectiveness of CBT-based interventions (including its

impact of secondary outcomes and potential moderators of the effect). The review also examines the acceptability of CBT interventions for PPD and adaptations to potentially improve acceptability of CBT for a PPD population. Conclusions will be drawn in the context of the limitations of the methodology adopted for this review. However, the review will provide a necessary step in addressing the current state of the CBT evidence base for PPD and results will have the potential to inform future PPD intervention developments in line with phase I of the Medical Research Council framework for developing complex interventions.[50]

## ETHICS AND DISSEMINATION PLAN

Systematic reviews potentially present two ethical challenges. First, informed consent provided for the original study is not necessarily valid for inclusion of data in the systematic review. Second, systematic reviews may include studies with ethical insufficiencies.[105] In addition, data will be extracted on ethical considerations for each included study with a summary paragraph included in the final manuscript under the heading 'Funding and ethics'.[105]

Dissemination to relevant audiences is a necessary step towards impact. This review will be of relevance to a range of audiences including those working in the peripartum, women's health and CBT fields. We will report review results through scientific conferences and publications, as well as publications for professional and lay audiences and meetings. We will also provide summaries of results and offer to present these for key parties including Health Insurance Directors and National and County Level Health Policy representatives. The authors will use social media and attend participatory involvement events to promote the review results and patient representatives will be contacted to feature results on social media, blogs and vlogs.

**Acknowledgements** Alkistis Skalkidou from the Department of Women's and Children's Health at Uppsala University for providing peer review of the search strategy. Agnes Kotka, Librarian at Uppsala University Library for assisting with the development of the electronic search strategy.

**Contributors** DP, JW and HO conceptualised the study. DP drafted the protocol. DP, JW, HO and ASS designed the study. DP, JW and HO assisted with manuscript writing. LvE, CA and OB participated critical revision of the study design and manuscript. All authors read and approved the final manuscript.

**Funding** This work was supported by U-CARE, which is a strategic research environment funded by the Swedish Research Council (dnr 2009–1093).

**Disclaimer** This funding source had no role in the design of this study and will not have any role during its execution, analyses, interpretation of the data or decision to submit results.

**Competing interests** None declared.

**Patient consent for publication** Not required.

**Ethics approval** This study protocol is for a systematic review, meta-analysis and meta sythesis using secondary data. No raw individual level data is included within this review, with only group level data extracted and analysed, as such no formal ethical approval is required by the National Ethical Review Board in Sweden.

**Provenance and peer review** Not commissioned; externally peer reviewed.

**ORCID iDs**
Danelle Pettman http://orcid.org/0000-0002-5956-4025
Heather O'Mahen http://orcid.org/0000-0003-3458-430X
Agneta Skoog Svanberg https://orcid.org/0000-0003-4729-9962
Louise von Essen https://orcid.org/0000-0001-5816-7231
Cathrine Axfors https://orcid.org/0000-0002-2706-1730
Joanne Woodford https://orcid.org/0000-0001-5062-6798

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
