## [Reviewer comments · BMJ Open]

ARTICLE DETAILS

TITLE (PROVISIONAL)	Effectiveness and acceptability of cognitive behavioural therapy based interventions for maternal peripartum depression: a systematic review, meta-analysis and thematic synthesis protocol.
AUTHORS	Pettman, Danelle; O'Mahen, Heather; Svanberg, Agneta; von Essen, Louise; Axfors, Cathrine; Blomberg, Oscar; Woodford, Joanne

VERSION 1 – REVIEW

REVIEWER	Ryan Van Lieshout McMaster University (Canada)
REVIEW RETURNED	05-Aug-2019

GENERAL COMMENTS	This is a nicely written and concise protocol that proposes to systematically review an important body of knowledge (CBT for peripartum depression) with implications for women, their children, their families, and healthcare systems around the world. My review is organized by manuscript section below: Abstract Since Sockol (Journal of Affective Disorders 177 (2015) 7–21) relatively recently published a systematic review and meta-analysis of CBT for treating and preventing perinatal depression, it is understandable that they state in the abstract that they wish to focus this piece on secondary outcomes (anxiety, etc). However, the protocol then states that depression is its primary focus. I suggest that the authors be more clear in the abstract about what the primary and secondary objectives of the proposed review are, and simply more clearly highlight its more novel aspects (secondary outcomes, moderators, qualitative outcomes). It is also not clear why Sockol's most recent systematic review (2015) on the topic they propose to review is not cited in the protocol. I recommend that they mention it and then make more explicit why this new systematic review and meta-analysis is warranted. Certainly their secondary outcomes, and a more extensive examination of potential moderating factors, along with their qualitative focus are all potentially novel, but they need to make a stronger case that the proposed work is not simply incremental in nature. Certainly, the research questions (of which there are multiple) that they propose are quite broad for a systematic review that would be published in a traditional biomedical journal (with its typical word count constraints). Have the authors considered conducting separate quantitative and qualitative reviews?
---

Introduction

This section is also nicely written and organized. The authors do a good job of describing peripartum depression, though there are important definitional issues that are not examined in as much depth as is ideal here. It could be of value to distinguish a bit more between antenatal and postpartum depression (etiology, presentation, course, impact on offspring), describing their differences and similarities, and then arguing in a more balanced way why these should be combined in a systematic review.

The prevalence rates that they describe also raise an important issue that deserves attention not only in this section, but in their methods. Rates of major depressive disorder in the postpartum period are generally cited as lower than the lower range of 10% that they describe. Moreover, rates closer to 25% have generally been found in studies using screening questionnaires (that have moderate specificity). While elevated rates of depressive symptoms may be important to treat, a discussion of definitional issues in the introductory section is potentially warranted, as is the conduct of analyses of moderators like MDD diagnosis (via structured interview or made by a clinician) vs. those arrived at using cut points relating to self-report questionnaires.

The protocol could also be strengthened by the provision of additional rationale for examining CBT, why they chose to examine CBT over IPT, and/or why they didn't simply look at both CBT and IPT. Certainly, combining the two would make for an even longer piece, but it could be helpful to articulate the rationale for the current focus rather than others.

Despite the authors' assertion in the abstract that a significant focus would be on three secondary outcomes (stress, parenting, anxiety), the rationale for this decision is not clearly described in this section (there really is only a single sentence on the topic).

While it is commendable that the authors wish to examine so many different outcomes (depression, anxiety, stress, parenting competence, parenting social support, parenting stress, individual stress, etc.) they should attempt to provide additional rationale for why these are all of value.

When they discuss their moderators, they should review those that Sockol examined in her most recent report (study design, control type, sample type, outcome measure, inclusion criteria) and describe why another review is warranted at this time.

There are also additional considerations below for the authors:

Reference 2 appears to relate to perinatal depression in fathers.

On line 20 of page 4: Authors can consider citing more than one study if they say 'recent studies'

On line 6 of page 5, it appears that ref 13 is not specific to women with peripartum depression and is rather looking at depression in adults in general. It may be beneficial for the reader to cite only studies specific for peripartum depression

In the second paragraph on page 5, it says that this is an updated systematic review, but previously it was mentioned that no research has been done on CBT's impact on perinatal anxiety/stress etc.

Suggest removing the word 'planned meta-analysis' in the second paragraph on page 6

Methods

The methods are generally well-written and the criteria (PICOS) are clearly outlined.

In terms of objectives, although data may not be widely available, it could make sense to also examine MDD and anxiety disorder diagnoses as outcomes in objectives 1 & 2. Or even the proportion of women in these trials that meet 'clinically significant improvement' cutoffs.

It is recommended that the authors outline their quantitative data extraction form more clearly (i.e., what exact participant characteristics will you be extracting?).

It is interesting that the authors want to exclude studies if they are deemed to have a high risk of bias (1st line on p.11). It may also be of value to include all eligible studies, regardless of risk of bias, and then conducting a sensitivity analysis (excluding studies with a high risk of bias) and seeing if the effect sizes change.

The funding constraints comment on p. 7 is a bit curious. They propose conducting quite an ambitious systematic review using a range of methods and software platforms and yet state that funding constraints prevented them from consulting women with postpartum depression. While such consultation may not be imperative to the conduct of the review, the rationale is a bit difficult to reconcile. In light of this, it would be of value to provide a bit more detail on how mothers with lived experience with PPD will be involved in the project moving forward.

The protocol could benefit from the provision of additional rationale for the use of a 16 year cutoff, as well as which EPDS cutoffs will be used and why (and if sensitivity analyses will be used for these different cut points). The authors may also wish to consider the impact of trial inclusion criteria (women with substance use problems, bipolar disorder, etc) on their results.

The provision of citations supporting the synthesis of traditional CBT treatment with problem solving and behavioural activation (Behavioural Therapy) would strengthen the protocol.

Some additional clarification would be helpful in the first full paragraph on page 9 where they suggest that they will allow active comparator treatments, but that CBT vs. Medication alone would not be eligible. Also, on page 9, it may be of value to state that structured clinical interview assessments are also a possible way for quantitative outcomes to be assessed. If these are included, they should describe the psychometric standards they will set for their inclusion.

On the second last line of page 10, they could consider listing some additional biases that occur in non-RCT designs.

On line 44 of page 11, the authors mention that no date restrictions apply, yet in the Introduction section you mentioned this would be 'an updated systematic review'. I recommend either putting a date restriction to truly make it updated, or removing the word 'updated'

from the Introduction section.

On line 51 of page 11, please describe in more detail what is meant by “forward citation searches”.

In terms of the secondary outcomes, will the same psychometric standards as with studies of depression apply? If so, they may wish to make that explicit.

Will the authors report statistics relating to agreement rates in their screening and data extraction processes.

It would be of value for the authors to clarify a bit more what they mean by “intention to treat principles will be followed as far as possible” (page 15, line 20-22).

I wonder if it might make sense if you chose the first follow-up time point to calculate your effect sizes and then use later end-points in your sensitivity analysis?

Alternatively, you may want to justify why you chose six months or earlier as your primary end-point.

Please remove the word ‘temporary’ from ‘temporary removal’ when discussing sensitivity analyses on page 15. I also recommend that you justify why you are conducting those three specific sensitivity analyses.

When talking about funnel asymmetry on page 15, please state the type of software you will use to assess for different types of biases.

In terms of moderators, baseline diagnoses, particularly comorbid anxiety disorders might be something to consider, especially given their relevance to response of anxiety symptoms to CBT targeted to depression.

For the qualitative extraction, I might consider listing the credentials of the two extractors as learning to code qualitative data and use NVivo requires some experience

Dissemination Plan

The authors should consider providing specific examples of conferences/meetings/publications that you wish to disseminate your work at.

Table 2

It may be helpful if rather than a single code for ‘exclusion’ you make the criteria more specific (e.g., 1) beyond 12 months postpartum, 2) no intervention, 3) not a CBT intervention etc.) That way when raters get together, they are easily able to identify why they (may) disagree

Searches

Searches are well done and take into account different regional spellings. Although, I recommend that they consider removal of the term ‘alexithymia’ as that is not synonymous with depression. The authors can also consider removing concept 4 (therapy terms) as this may unnecessarily restrict your search for all types of CBT interventions (and this is important to you as delivery type is a moderator you’re interested in looking at). I can appreciate that this

	may reduce specificity (as you outlined on p.4 of the PRESS Guideline) although it may be better to be as inclusive as possible rather than exclusive if you want to ensure you capture all possible articles Summary: This ambitious protocol aims to address both quantitative and qualitative outcomes in CBT for peripartum depression in the form of a meta-analysis and thematic synthesis. While the methods are largely sound, additional rationale are required in the introductory section, and the authors do need to provide stronger arguments for why an update of Sockol's 2015 review is required at this point in time. The provision of more compelling arguments will highlight why this work is not simply just incremental. Moreover, it is important that the authors provide a more fulsome discussion of the secondary outcomes, their importance, and their amenability to treatment.
--	---

REVIEWER	Fiona Challacombe King's College London, UK
REVIEW RETURNED	12-Sep-2019

GENERAL COMMENTS	This is a very clearly written and laid out protocol for what is likely to be an important and influential evidence review. The strategy is very clear and comprehensive. (nb: minor point - references 7 and 8 are the same).
--

VERSION 1 – AUTHOR RESPONSE

Reviewer 1:

Abstract

3. Since Sockol (Journal of Affective Disorders 177 (2015) 7–21) relatively recently published a systematic review and meta-analysis of CBT for treating and preventing perinatal depression, it is understandable that they state in the abstract that they wish to focus this piece on secondary outcomes (anxiety, etc). However, the protocol then states that depression is its primary focus. I suggest that the authors be more clear in the abstract about what the primary and secondary objectives of the proposed review are, and simply more clearly highlight its more novel aspects (secondary outcomes, moderators, qualitative outcomes).

Thank you for highlighting this discrepancy. The primary and secondary outcomes have been made clearer in the abstract:

Page 2:

The primary aim of this review is to examine the overall effectiveness of CBT based interventions for peripartum depression. Secondary aims are to explore the effect of CBT based interventions targeted at peripartum depression on novel secondary outcomes and moderators potentially associated with effectiveness. To date, there has been little examination of effect on important secondary outcomes (e.g., anxiety, stress and parenting), nor clinical and methodological moderators. Further, this review aims to explore the acceptability of CBT based interventions for women with peripartum depression and examine important adaptations for this population.

4. It is also not clear why Sockol's most recent systematic review (2015) on the topic they propose to

review is not cited in the protocol. I recommend that they mention it and then make more explicit why this new systematic review and meta-analysis is warranted. Certainly their secondary outcomes, and a more extensive examination of potential moderating factors, along with their qualitative focus are all potentially novel, but they need to make a stronger case that the proposed work is not simply incremental in nature.

Thank you for highlighting this oversight with the Sockol (2015) paper. The reference has been added and a fuller explanation of why this new systematic review and meta-analysis is warranted is given;

Page 6-7:

A further omission in the current evidence base of interventions for PPD concerns clinical and methodological moderators potentially associated with effectiveness. Whilst a recent review has explored the effect of a variety of moderators on CBT for PPD, including type of study control condition (e.g., wait-list control or treatment-as-usual) and time point of intervention (e.g., antepartum or postpartum)[18] a number of potentially important moderators were not examined. As such, the present review will facilitate the exploration of a number of novel moderators for example the potential effect of health professional delivering intervention (e.g., psychologists, family doctors, nurses and non-professionals). Investigating the potential impact of these moderators is of importance given CBT based interventions are delivered by a wide range of health professionals. [31] In addition, we will examine the potential effect of including interventions that also include components targeting parenting. This is of particular importance given that research has consistently associated parenting difficulties with postnatal depression.[44] However, to date, to the best of our knowledge, existing reviews have not examined the inclusion of parenting components within CBT based interventions as a moderator. Increased understanding of clinical moderators associated with effectiveness are particularly important to inform the development of future interventions for the population.

Refs:

18. Sockol LE. A systematic review of the efficacy of cognitive behavioral therapy for treating and preventing perinatal depression. *J Affect Disord* 2015;177:7–21. doi:10.1016/j.jad.2015.01.052

31. Stamou G, García-palacios A, Botella C. Cognitive-Behavioural therapy and interpersonal psychotherapy for the treatment of post-natal depression : a narrative review. *BMC Psychol* 2018;18. doi:10.1186/s40359-018-0240-5

44. Murray L, Cooper P, Fearon P. Parenting difficulties and postnatal depression: Implications for primary healthcare assessment and intervention. *Community Pract* 2014;87:34–8.

5. Certainly, the research questions (of which there are multiple) that they propose are quite broad for a systematic review that would be published in a traditional biomedical journal (with its typical word count constraints). Have the authors considered conducting separate quantitative and qualitative reviews?

Thank-you for this important comment. We are currently aiming to submit to a journal specializing in reviews in the area of clinical psychology (e.g. *Clinical Psychology Review*) and thus have higher word count than traditional medical journals. We aim to report the quantitative and qualitative data together in order to triangulate findings. However, should reporting the qualitative and quantitative results together become problematic due to word count, we can look to report the findings separately.

Introduction

6. This section is also nicely written and organized. The authors do a good job of describing peripartum depression, though there are important definitional issues that are not examined in as much depth as is ideal here. It could be of value to distinguish a bit more between antenatal and postpartum depression (etiology, presentation, course, impact on offspring), describing their differences and similarities, and then arguing in a more balanced way why these should be combined in a systematic review.

Thank you for highlighting this omission. A fuller discussion of antenatal and postnatal depression is given below:

Page 4:

Peripartum depression (PPD) is a common mental health difficulty characterised by low mood during pregnancy (antepartum period) and/or after childbirth (postpartum period).[1] Whilst it has been argued depression occurring during pregnancy is distinctly different to postpartum depression [2] in terms of epidemiological, [3] hormonal sensitivity [4] and immune system functioning [5] for the purposes of this review we will use the more inclusive “peripartum” definition and examine time point of intervention (e.g., antepartum or postpartum period) as a potential moderator of effect.

Refs:

1. American Psychiatric Association. Diagnostic and statistical manual of mental disorders 5th edition: DSM-V. Washington DC: American Psychaitric Association 2013.
2. Payne JL. Recent Advances and Controversies in Peripartum Depression. *Curr Obstet Gynecol Rep* 2016;5:250–6. doi:10.1007/s13669-016-0167-x
3. Vesga-López O, Blanco C, Keyes K, et al. Psychiatric disorders in pregnant and postpartum women in the United States. *Arch Gen Psychiatry* 2008;65:805–15. doi:10.1001/archpsyc.65.7.805
4. Bloch M, Schmidt PJ, Danaceau M, et al. Effects of gonadal steroids in women with a history of postpartum depression. *Am J Psychiatry* 2000;157:924–30. doi:10.1176/appi.ajp.157.6.924
5. Sharma V, Mazmanian D. The DSM-5 peripartum specifier: Prospects and pitfalls. *Arch Womens Ment Health* 2014;17:171–3. doi:10.1007/s00737-013-0406-3

7. The prevalence rates that they describe also raise an important issue that deserves attention not only in this section, but in their methods. Rates of major depressive disorder in the postpartum period are generally cited as lower than the lower range of 10% that they describe. Moreover, rates closer to 25% have generally been found in studies using screening questionnaires (that have moderate specificity). While elevated rates of depressive symptoms may be important to treat, a discussion of definitional issues in the introductory section is potentially warranted, as is the conduct of analyses of moderators like MDD diagnosis (via structured interview or made by a clinician) vs. those arrived at using cut points relating to self-report questionnaires.

Thank-you for these important points. First, we have now included in the Introduction information pertaining to different prevalence rates dependent on the use of diagnostic or screening tools only:

Page 4:

Prevalence rates of PPD vary dependent on whether measured via screening scales or standardised diagnostic tools, [6] with the former at a rate of 19.2% and the latter with at a rate of 14.5%.[7] A recent study indicates symptoms of depression are more prevalent during the postpartum period than in age matched controls.[8]

Second, we will now include ‘Diagnosis of major depressive disorder (yes/no)’ as a moderator. Please note the data extraction section under Methods (added from comment #17) and the Data Extraction Form (supplementary appendix 5) have been amended to include the new moderator for major depression:

2. Participant characteristics; method of depression assessment (e.g., structured clinical interview or screening tool), major depression diagnosis (yes/no).

Refs:

6. Misri S, Abizadeh J, Nirwin S. Depression during pregnancy and postpartum period. In: Wenzel A, ed. *The Oxford Handbook of Perinatal Psychology*. Oxford University Press 2016. 111–31.
7. Gavin NI, Gaynes BN, Lorh KN, et al. Perinatal depression: a systematic review of prevalence and incidence. *Obstet Gynecol* 2005;106:1071–83. doi:10.1097/01.AOG.0000183597.31630.db
8. Merkitich KG, Jonas KG, O'Hara MW. Modeling trait depression amplifies the effect of childbearing on postpartum depression. *J Affect Disord* 2017;223:69–75. doi:10.1016/j.jad.2017.07.017

8. The protocol could also be strengthened by the provision of additional rationale for examining CBT, why they chose to examine CBT over IPT, and/or why they didn't simply look at both CBT and IPT. Certainly, combining the two would make for an even longer piece, but it could be helpful to articulate the rationale for the current focus rather than others.

Additional rationale for examining CBT provided below:

Page 5:

Although there are mixed findings regarding the superiority of one type of psychological intervention over another,[17,19,27] several reviews have found CBT to be an effective psychological intervention for PPD.[18,25,30,31]. As such CBT based interventions are recommended as a first-line intervention in clinical guidelines for a number of common mental health difficulties.[32] Additionally, in comparison with other therapeutic approaches, CBT is provided in a wider range of intervention formats including; individual, [33] group [34] and internet-administered. [35].

Refs:

32. David D, Cristea I, Hofmann SG. Why Cognitive Behavioral Therapy is the Current Gold Standard of psychotherapy. *Front Psychiatry* 2018;9:10–3. doi:10.3389/fpsy.2018.00004
33. O'Mahen H, Himle JA, Fedock G, et al. A pilot randomized controlled trial of cognitive behavioral therapy for perinatal depression adapted for women with low incomes. *Depress Anxiety* 2013;30:679–87. doi:http://0-dx.doi.org.lib.exeter.ac.uk/10.1002/da.22050
34. Van Lieshout RJ, Yang L, Haber E, et al. Evaluating the effectiveness of a brief group cognitive behavioural therapy intervention for perinatal depression. *Arch Womens Ment Health* 2017;20:225–8. doi:10.1007/s00737-016-0666-9
35. Lau Y, Htun TP, Wong SN, et al. Therapist-supported internet-based cognitive behavior therapy for stress, anxiety, and depressive symptoms among postpartum women: A systematic review and meta-analysis. *J Med Internet Res* 2017;19:1–17. doi:10.2196/jmir.6712

9. Despite the authors assertion in the abstract that a significant focus would be on three secondary outcomes (stress, parenting, anxiety), the rationale for this decision is not clearly described in this section (there really is only a single sentence on the topic).

While it is commendable that the authors wish to examine so many different outcomes (depression, anxiety, stress, parenting competence, parenting social support, parenting stress, individual stress, etc.) they should attempt to provide additional rationale for why these are all of value.

Additional rationale regarding secondary outcomes has been provided below

Pages 5-6:

Whilst existing interventions have demonstrated CBT based interventions for PPD to be effective for depression they have often neglected important secondary outcomes e.g., anxiety, stress (individual and perceived parenting), parenting (e.g., sensitivity/responsiveness) perceived social support and perceived parental competence has been largely unexamined. It is important to examine the effect of interventions for PPD on symptoms of anxiety given comorbidity rates [36] and research suggesting both maternal anxiety and depression should be addressed in interventions.[37] An examination of effect on stress (individual and perceived parenting) is also warranted given the deleterious impact of stress on mothers and infants.[38] Indeed, perceived parenting stress disrupts the ability of mothers to appropriately assess infant signals and to react sensitively to them.[39] Further, research suggests interventions for PPD have little benefit on child outcomes,[40] and therefore both maternal depression and parenting difficulties should be addressed.[41] Additionally, perceived low levels of parental competence.[42] and poor social support are associated with PPD suggesting interventions should aim to improve both and be included as a secondary outcome.[43]

Refs:

36. Biaggi A, Conroy S, Pawlby S, et al. Identifying the women at risk of antenatal anxiety and depression: A systematic review. *J. Affect. Disord.* 2016;191:62–77. doi:10.1016/j.jad.2015.11.014
37. Barker ED, Jaffee SR, Uher R, et al. The contribution of prenatal and postnatal maternal anxiety and depression to child maladjustment. *Depress Anxiety* 2011;28.
38. Dunkel Schetter C, Tanner L. Anxiety, depression and stress in pregnancy: Implications for mothers, children, research, and practice. *Curr. Opin. Psychiatry.* 2012;25:141–8. doi:10.1097/YCO.0b013e3283503680
39. Mills-Koonce WR, Appleyard K, Barnett M, et al. Adult attachment style and stress as risk factors for early maternal sensitivity and negativity. *Infant Ment Health J* 2011;32:277–85. doi:10.1002/imhj.20296
40. O’Hara MW. Postpartum depression: what we know. *J Clin Psychol* 2009;65:1258–69. doi:10.1002/jclp.20644
41. Stein A, Pearson RM, Goodman SH, et al. Effects of perinatal mental disorders on the fetus and child. *Lancet (London, England)* 2014;384:1800–19. doi:10.1016/S0140-6736(14)61277-0
42. Takács L, Smolík F, Putnam S. Assessing longitudinal pathways between maternal depressive symptoms, parenting self-esteem and infant temperament. *PLoS One* 2019;14:e0220633. doi:10.1371/journal.pone.0220633
43. Tambag H, Turan Z, Tolun S, et al. Perceived social support and depression levels of women in the postpartum period in Hatay, Turkey. *Niger J Clin Pract* 2018;21:1525–30. doi:10.4103/njcp.njcp_285_17

10. When they discuss their moderators, they should review those that Sockol examined in her most recent report (study design, control type, sample type, outcome measure, inclusion criteria) and describe why another review is warranted at this time.

This point has been addressed in response to comment #4 relating to the Sockol (2015) review.

11. Reference 2 appears to relate to perinatal depression in fathers.

Thank-you for highlighting this error. This reference has been removed.

12. On line 20 of page 4: Authors can consider citing more than one study if they say 'recent studies'

This has been corrected

Page 4:

A recent study indicates symptoms of depression are more prevalent during the postpartum period than in age matched controls

13. On line 6 of page 5, it appears that ref 13 is not specific to women with peripartum depression and is rather looking at depression in adults in general. It may be beneficial for the reader to cite only studies specific for peripartum depression

The paper referring to general depression has been removed from this section.

14. In the second paragraph on page 5, it says that this is an updated systematic review, but previously it was mentioned that no research has been done on CBT's impact on perinatal anxiety/stress etc.

This term has been removed.

15. Suggest removing the word 'planned meta-analysis' in the second paragraph on page 6

This term has been removed

Methods

16. The methods are generally well-written and the criteria (PICOS) are clearly outlined.

In terms of objectives, although data may not be widely available, it could make sense to also examine MDD and anxiety disorder diagnoses as outcomes in objectives 1 & 2. Or even the proportion of women in these trials that meet 'clinically significant improvement' cutoffs.

Thank-you for this important comment. As highlighted in #7 we will now include Diagnosis of major depressive disorder (yes/no) as a moderator. Whilst we also recognise that diagnosis of anxiety disorders is also very important in the population, as anxiety is a secondary outcome in the present review, we have decided not to include diagnosis of anxiety disorders as a moderator. However, we have added to the data extraction form a section to extract data concerning comorbid diagnoses and we will report these under Participant Characteristics in the manuscript when reporting the results of the review.

In addition, we have considered your comment concerning examining the proportion of women in these trials that meet 'clinically significant improvement' cut-offs concerning depression and anxiety in detail. We have examined some of the wider literature concerning scale derived cut offs. Difficulties with using this approach include that measures often have several different cut offs (we will likely be assessing several different depression measures) and if cut offs are chosen post-hoc they are potentially an inappropriate manipulation of the data (Leucht et al., 2007). Additionally, dichotomizing continuous variables is discouraged (Dawson & Weiss, 2012). Therefore, we decided not to report the proportion of women that meet 'clinically significant improvement' via cut-offs within the present review.

Refs:

Leucht S, Davis JM, Engel RR, et al. Defining 'response' in antipsychotic drug trials: Recommendations for the use of scale-derived cutoffs. *Neuropsychopharmacology* 2007;32:1903–10. doi:10.1038/sj.npp.1301325

Dawson N V., Weiss R. Dichotomizing continuous variables in statistical analysis: A practice to avoid. *Med Decis Mak* 2012;32:225–6. doi:10.1177/0272989X12437605

17. It is recommended that the authors outline their quantitative data extraction form more clearly (i.e., what exact participant characteristics will you be extracting?).

More details to clarify the exact data being extracted has now been provided in the main manuscript text:

Page 16-18:

The following data will be extracted:

1. Study characteristics; citation, publication type (published or unpublished), country of origin, funding source, language, aims and objectives, design and inclusion/exclusion criteria.

2. Participant characteristics; method of depression assessment (e.g., structured clinical interview or screening tool), major depression diagnosis (yes/no), recruitment setting (clinical or community), age of mother (Mean, SD), age of infant (Mean, SD), time point of intervention (e.g., antepartum or postpartum), baseline anxiety, comorbidities, ethnicity of mother (n, %), relationship status (n, %), educational status (n, %), employment status (n, %), mothers first child (n, %), average household income (n, %), breastfeeding (n, %), level of depression severity at baseline (if clinical cut offs available), depression chronicity years (Mean, SD), number of participants invited, number of participants screened, number of participants eligible, number of participants randomised and reasons for non-eligibility.

3. Intervention characteristics; type of CBT intervention (CBT, BA or problem solving), inclusion of social components (e.g., peer support group or involvement of partner; yes/no), inclusion of parenting intervention components (e.g., video interaction guidance; yes/no), treatment manual (yes/no), measurement of treatment adherence (yes/no), method of delivery (e.g., face to face, group or internet-administered), treatment setting (e.g., clinic or community), health professional delivering intervention (e.g., clinical psychologist or midwife), study specific training (yes/no), duration of treatment (weeks), number of sessions, length of sessions (minutes), maximum length of treatment sessions over treatment course (minutes), group size (Mean, SD) and type of control condition used (e.g., no-treatment control, wait-list control, TAU, non-specific factors component control, specific factors component control and active comparator).

4. Outcome measurements; Primary outcome: outcome measure used, participant n, Mean, SD and/or SE for each outcome time point collected and measure of outcome quality (Cronbach alpha for internal consistency and test retest reliability from original validation papers). Secondary outcome(s) (including; anxiety, individual stress, perceived parental stress, self-report parenting, perceived social support, parental competence and observational parenting): outcome measure used, participant n, Mean, SD and/or SE for each outcome time point collected.

5. Statistical techniques; power calculation, intention to treat (yes/no), method of dealing with missing data, baseline comparability and for cluster trials, estimates of intra-cluster correlation coefficients (ICC) will be gathered).

6. Participant flow; randomised to intervention, randomised to control, lost to follow-up (at each time point measured), analysed intervention (at each time point measured) and analysed control (at each time point measured) and attrition rate.

7. Research ethics; data relating to ethics (e.g., ethical approval, ethical issues highlighted).

18. It is interesting that the authors want to exclude studies if they are deemed to have a high risk of bias (1st line on p.11). It may also be of value to include all eligible studies, regardless of risk of bias, and then conducting a sensitivity analysis (excluding studies with a high risk of bias) and seeing if the effect sizes change.

Excluding studies with high risk of selection bias is a method previously adopted in a number of other published systematic reviews and meta-analyses (Farrand & Woodford, 2013; 2015; Woodford et al., 2013) and has been positively commented on in previous review processes. The reduction of selection bias by the use of adequate randomisation procedures to avoid over inflation of effect sizes (Gellatly et al., 2007). However, other risk of biases will be reported on, using the Cochrane risk-of-bias tool for randomized trials (RoB 2; Higgins et al., 2018) and included in the sensitively analysis. Further justification has been added to the manuscript:

Page 13:

Further, RCTs with randomisation procedures at allocation and concealment rated as high risk of bias, in line with the revised Cochrane risk-of-bias tool for randomized trials (RoB 2), [61] will be excluded (see online supplementary appendix 2).

This is to minimise the inclusion of studies of low quality with high risk of selection bias known to inflate effect sizes (for example,[80,81]) and is a technique used in previous systematic reviews and meta-analyses.[82–84]

Refs:

80. Gellatly J, Bower P, Hennessy S, et al. What makes self-help interventions effective in the management of depressive symptoms? Meta-analysis and meta-regression. *Psychol. Med.* 2007;37:1217–28. doi:10.1017/S0033291707000062

81. Cuijpers P, van Straten A, Bohlmeijer E, et al. The effects of psychotherapy for adult depression are overestimated: a meta-analysis of study quality and effect size. *Psychol Med* 2010;40:211–23. doi:10.1017/S0033291709006114

82. Farrand P, Woodford J. Impact of support on the effectiveness of written cognitive behavioural self-help: A systematic review and meta-analysis of randomised controlled trials. *Clin Psychol Rev* 2013;33:182–95. doi:10.1016/j.cpr.2012.11.001

83. Farrand P, Woodford J. Effectiveness of cognitive behavioural self-help for the treatment of depression and anxiety in people with long-term physical health conditions: a systematic review and meta-analysis of randomised controlled trials. *Ann Behav Med A Publ Soc Behav Med* 2015;49:579–93. doi:10.1007/s12160-015-9689-0

84. Woodford J, Farrand P, Richards D, et al. Psychological treatments for common mental health problems experienced by informal carers of adults with chronic physical health conditions (Protocol). *Syst Rev* 2013;2:9. doi:10.1186/2046-4053-2-9

19. The funding constraints comment on p. 7 is a bit curious. They propose conducting quite an ambitious systematic review using a range of methods and software platforms and yet state that funding constraints prevented them from consulting women with postpartum depression. While such consultation may not be imperative to the conduct of the review, the rationale is a bit difficult to

reconcile. In light of this, it would be of value to provide a bit more detail on how mothers with lived experience with PPD will be involved in the project moving forward.

With regards to funding the research group has access to software via the university, however the cost of involving mothers with lived experience further in this study (including their expenses and travel) was deemed out of scope for the systematic review. However, we plan to involve mothers to help interpret the

qualitative synthesis, the second study in this PhD project will be a participatory action research design involving mothers with lived experience to develop a new psychological intervention. This has now been highlighted in the manuscript as below:

Page 9:

Patient and public involvement:

Due to project funding constraints patients and members of the public were not involved in the design of this protocol. However, results of the present review will be used to inform the further development of a PPD intervention, following the MRC complex interventions framework [52] targeting both depression and parenting. We will adopt a participatory action research approach [53] to inform the development of the intervention, placing mothers with lived experience of PPD at the centre of the intervention development process.

Refs:

- 52. Craig P, Dieppe P, Macintyre S, et al. Developing and evaluating complex interventions: The new Medical Research Council guidance. *Bmj* 2008;337:979–83. doi:10.1136/bmj.a1655
53. Baum F, MacDougall C, Smith D. Participatory action research. *J Epidemiol Community Health* 2006;60:854–7. doi:10.1136/jech.2004.028662

20. The protocol could benefit from the provision of additional rationale for the use of a 16 year cutoff, as well as which EPDS cutoffs will be used and why (and if sensitivity analyses will be used for these different cut points). The authors may also wish to consider the impact of trial inclusion criteria (women with substance use problems, bipolar disorder, etc) on their results.

The following rationale for the 16 year cut off was added to the protocol:

Pages 9-10:

Participants. Adult women aged ≥ 16 years (a cut-off of 16 years was chosen as studies are likely to be drawn from international settings with variation in the age at which someone is deemed an adult).[55,56]

Refs.

55. European Union Agency for Fundamental Rights. Age of majority.

<https://fra.europa.eu/en/publication/2017/mapping-minimum-age-requirements/age-majority> (accessed 10 Oct 2019).

56. US Legal. Age of Majority. <https://minors.uslegal.com/age-of-majority/> (accessed 10 Oct 2019).

Thank you for this comment regarding the use of EPDS cut-offs for study inclusion. We now realise the manuscript was unclear as we will not use EPDS cut-offs for study inclusion as no limitations will be placed on depression severity. We have therefore increased the clarity of our inclusion criteria in the PICOS as follows:

Page 10 and Additional file 2 – PICOS statement:

From pregnancy to 12 months postpartum with either; (1) a diagnosis of PPD e.g., Diagnostic and Statistical Manual of Mental Disorders (DSM) IV or V [1,57]; or (2) reporting depression symptomatology within the peripartum period (from pregnancy to 1 year postpartum) using a validated tool e.g., Edinburgh Postnatal Depression Scale (EPDS;[58]). **No limits will be placed on the severity of depression at baseline.** Exclusion criteria are; (1) intervention for mood disorders other than depression (e.g., bipolar affective disorder); and

(2) intervention explicitly focussed on targeting the prevention of maternal psychopathology in at-risk mothers.[25] For example, a study with the aim to treat current depression during the antepartum period,

with an aim to prevent depression during the postpartum period would be eligible for inclusion. However, interventions explicitly targeting the prevention of depression during either the antepartum or postpartum period will be excluded.

Refs.

1. American Psychiatric Association. Diagnostic and statistical manual of mental disorders 5th edition: DSM-V. Washington DC: American Psychaitric Association 2013.

57. American Psychiatric Association. Diagnostic and statistical manual of mental disorders 4th edition : DSM-IV. 1994.

58. Cox JL, Holden JM, Sagovsky R. Detection of postnatal depression. Development of the 10-item Edinburgh Postnatal Depression Scale. Br J Psychiatry J Ment Sci 1987;150:782–

6. <http://www.ncbi.nlm.nih.gov/pubmed/3651732>

25. Nillni YI, Mehralizade A, Mayer L, et al. Treatment of depression, anxiety, and trauma-related disorders during the perinatal period: A systematic review. Clin Psychol Rev Published Online First: 2018. doi:10.1016/j.cpr.2018.06.004

21. The provision of citations supporting the synthesis of traditional CBT treatment with problem solving and behavioural activation (Behavioural Therapy) would strengthen the protocol.

A citation supporting the synthesis of CBT treatment with problem solving and behavioural activation has been added to the protocol:

Page 10:

Eligible interventions will state the use of CBT based interventions, including behavioural activation or problem-solving, and contain specific therapeutic components associated with these approaches. These approaches have been previously synthesised in a meta-analysis for psychotherapy in major depression.[17]

Ref:

17. Cuijpers P, van Straten A, Andersson G, et al. Psychotherapy for depression in adults: A meta-analysis of comparative outcome studies. J Consult Clin Psychol 2008;76:909–22. doi:10.1037/a0013075

22. Some additional clarification would be helpful in the first full paragraph on page 9 where they suggest that they will allow active comparator treatments, but that CBT vs. Medication alone would not be eligible.

Additional clarification regarding the types of active comparator that will be eligible is provide below:

Page 11:

Only trial designs that allow for the isolation of the effects of CBT will be included given it is important for active comparators to discriminate intervention effects.[61] For example, a study comparing CBT alone versus medication alone would be excluded as it would not be possible to isolate the effect of the CBT. However, CBT plus medication versus medication alone and CBT plus information versus information alone would be eligible for inclusion.

Ref.

61. Evans SR. Fundamentals of clinical trial design. J Exp Stroke Transl Med 2010;3:19–27

23. Also, on page 9, it may be of value to state that structured clinical interview assessments are also a possible way for quantitative outcomes to be assessed. If these are included, they should describe the psychometric standards they will set for their inclusion.

Details of how structured clinical interviews will be incorporated into the review have been added to the manuscript:

Page 12:

Quantitative outcomes. Studies eligible for inclusion will use a self-report or proxy/clinician administered standardised measurement of depression (e.g., the Beck Depression Inventory);[63] or PPD (e.g., EPDS) [58]. In order to ensure the quality of the outcome measurements concerning the primary outcome of depression only studies using depression outcome measurements with at least acceptable internal consistency and test-retest reliability at Cronbach's alpha ≥ 0.70 (as reported in outcome measurement validation studies) will be included.[64] In addition the proportion of mothers meeting diagnostic criteria for depression following intervention will be assessed. In order to ensure quality, only diagnosis made with either the SCID-I [65] or Mini-International Neuropsychiatric Interview (MINI) [66] will be included, as a recent systematic review has highlighted that only these two tools meet sensitivity and specificity criteria.[67]

Refs.

65. First M, Williams J, Karg R, et al. Structured Clinical Interview for DSM-5 (SCID-5 for DSM-5, Research Version; SCID-5-RV). Am Psychiatr Assoc
66. Sheehan D V., Lecrubier Y, Sheehan KH, et al. The Mini-International Neuropsychiatric Interview (M.I.N.I.): The development and validation of a structured diagnostic psychiatric interview for DSM-IV and ICD-10. In: Journal of Clinical Psychiatry. 1998. 22–33.
67. Pettersson A, Boström KB, Gustavsson P, et al. Which instruments to support diagnosis of depression have sufficient accuracy? A systematic review. Nord J Psychiatry 2015;69:497–508. doi:10.3109/08039488.2015.1008568

24. On the second last line of page 10, they could consider listing some additional biases that occur in non-RCT designs.

Details about additional biases of non-RCT trials have been added:

Pages 13:

RCT designs were chosen for this systematic review as they are less prone to biases inherent with other study designs.[76] Randomisation aims to balance known and unknown variables

between the treatment groups in order to control for confounding. Random allocation also facilitates the blinding of interventions e.g., to evaluators.[77] Meta-epidemiological studies have shown that trials without proper RCT design, i.e., with inadequate concealment of treatment allocation or inadequate randomization, tend to report higher effect sizes.[78]

Refs:

76. Hariton E, Locascio JJ. Randomised controlled trials - the gold standard for effectiveness research: Study design: randomised controlled trials. *BJOG* 2018;125:1716. doi:10.1111/1471-0528.15199

77. Moher D, Hopewell S, Schulz KF, et al. CONSORT 2010 explanation and elaboration: Updated guidelines for reporting parallel group randomised trials. *Int J Surg* 2012;10:28–55. doi:10.1016/j.ijsu.2011.10.001

78. Dechartres A, Trinquart L, Faber T, et al. Empirical evaluation of which trial characteristics are associated with treatment effect estimates. *J Clin Epidemiol* 2016;77:24–37.

doi:10.1016/j.jclinepi.2016.04.005

25. On line 44 of page 11, the authors mention that no date restrictions apply, yet in the Introduction section you mentioned this would be ‘an updated systematic review’. I recommend either putting a date restriction to truly make it updated, or removing the word ‘updated’ from the Introduction section.

Reference to “an updated systematic review” has been removed from the introduction.

26. On line 51 of page 11, please describe in more detail what is meant by “forward citation searches”.

More detail has been added:

Page 14:

Forward citation searches using Google Scholar (forward citation chasing) [86] will be conducted for all included studies and reference lists of all of the included studies will be hand searched.

Ref:

86. Cooper C, Booth A, Varley-Campbell J, et al. Defining the process to literature searching in systematic reviews: A literature review of guidance and supporting studies. *BMC Med. Res. Methodol.* 2018;18. doi:10.1186/s12874-018-0545-3

27. In terms of the secondary outcomes, will the same psychometric standards as with studies of depression apply? If so, they may wish to make that explicit.

The psychometric properties of the secondary outcomes will be reported where available to aid interpretation. However exclusions will not be based on the basis of quality as this piece is exploratory.

28. Will the authors report statistics relating to agreement rates in their screening and data extraction processes.

The reporting of agreement statistics (kappa) is not recommended by Cochrane. “Comparison of a value of kappa with arbitrary cut-points is unlikely to convey the real impact of any disagreements on the review. For example, disagreement about the eligibility of a large, well conducted, study will have more substantial implications for the review than disagreement about a small study with risks of bias. The reasons for any disagreement should be explored. They may reveal the need to revisit eligibility criteria or coding schemes for data collection, and any resulting changes should be reported” (Higgins & Green, 2011). We will instead narratively describe any discrepancies if they arise during the selection process and document any changes to the eligibility criteria and/or coding schemes.

Ref: Higgins J., Green S. Cochrane handbook for systematic reviews of interventions version 5.1.0. Cochrane Collab 2011

29. It would be of value for the authors to clarify a bit more what they mean by “intention to treat principles will be followed as far as possible” (page 15, line 20-22).

Clarification of the intention to treat principle has been added to the manuscript:

Page 20:

Intention-to-treat principles stipulate the inclusion of every subject who is randomized according to randomized intervention assignment [95] and will be followed as far as possible. In instances in which intention-to-treat data is not available completer data will be extracted.

Ref:

95. Gupta S. Intention-to-treat concept: A review. Perspect Clin Res 2011;2:109. doi:10.4103/2229-3485.83221

30. I wonder if it might make sense if you chose the first follow-up time point to calculate your effect sizes and then use later end-points in your sensitivity analysis?

Alternatively, you may want to justify why you chose six months or earlier as your primary end-point.

A number of meta-analyses of psychological interventions have demonstrated elevated effect sizes at short-term/post-treatment follow-up in comparison with longer-term outcomes (Ekers et al., 2014; Farrand & Woodford, 2013; Flückiger et al., 2014). Therefore, outcomes taken at less than or equal to 6 months have been determined a priori as the primary end-point for the meta-analysis to reduce the chance of reporting an elevated treatment effect associated with focusing on post-treatment outcomes. The effect size for different lengths of follow-up (including post-treatment) will however be examined as length of follow-up is a pre-specified moderator within the meta-analysis. This has been clarified in the manuscript:

Page 19:

Where multiple time points are reported, a primary time point of ≤ 6 months post-treatment will be adopted for the primary analysis to reduce the likelihood of bias associated with examining short-term post-treatment effects only that are likely to result in elevated effect sizes. [82,92,93] However, length of follow-up will be included as a moderator.

Refs:

82. Farrand P, Woodford J. Impact of support on the effectiveness of written cognitive behavioural self-help: A systematic review and meta-analysis of randomised controlled trials. *Clin Psychol Rev* 2013;33:182–95. doi:10.1016/j.cpr.2012.11.001

92. Ekers D, Webster L, Van Straten A, et al. Behavioural activation for depression; An update of meta-analysis of effectiveness and sub group analysis. *PLoS One* 2014;9. doi:10.1371/journal.pone.0100100

93. Flückiger C, Del Re AC, Munder T, et al. Enduring effects of evidence-based psychotherapies in acute

depression and anxiety disorders versus treatment as usual at follow-up - A longitudinal meta-analysis. *Clin. Psychol. Rev.* 2014;34:367–75. doi:10.1016/j.cpr.2014.05.001

31. Please remove the word 'temporary' from 'temporary removal' when discussing sensitivity analyses on page 15. I also recommend that you justify why you are conducting those three specific sensitivity analyses.

The word 'temporary' has been removed and justification of the sensitivity analysis has been added;

Page 20-21:

Sensitivity analysis. A sensitivity analysis regarding the overall effect size of the primary outcome (depression) will be conducted by removal of (1) each study individually from the overall analysis and the effect size recalculated in order to estimate the statistical validity of the effect size; [98] (2) small studies ($n \leq 20$ across conditions) to explore the suggestion that small trials tend to report larger treatment benefits than larger trials [99]; and/or (3) studies with high attrition rates ($\geq 30\%$ in at least one arm), given attrition rates of $\geq 30\%$ are associated with imbalances at baseline and are therefore vulnerable to bias such as clinical and psychosocial differences between groups due to differences between those that leave or remain in the study, which is likely to have an impact upon the estimated effect of the exposure. [100]

Refs:

98. Willis BH, Riley RD. Measuring the statistical validity of summary meta-analysis and meta-regression results for use in clinical practice. *Stat Med* 2017;36:3283–301.

doi:10.1002/sim.7372

99. Greco T, Zangrillo A, Biondi-Zoccai G, et al. Meta-analysis: pitfalls and hints. *Hear lung Vessel* 2013;5:219–

25. doi:10.4028/www.scientific.net/AMR.60-61.110

100. Nunan D, Aronson J, Bankhead C. Catalogue of bias: attrition bias. *BMJ evidence-based Med* 2018;23:21–2. doi:10.1136/ebmed-2017-110883

32. When talking about funnel asymmetry on page 15, please state the type of software you will use to assess for different types of biases.

The software to be used was added

Page 21:

Comprehensive Meta-analysis (version 2) will be used to assess these different types of bias

33. In terms of moderators, baseline diagnoses, particularly comorbid anxiety disorders might be something to consider, especially given their relevance to response of anxiety symptoms to CBT targeted to depression.

Thank you for highlighting this. We have added baseline depression diagnosis but will not include baseline anxiety as a moderator (as addressed in comment #16). We have however, added baseline anxiety and comorbidities to the participant characteristics collected and these will be discussed in the clinical implications of the review paper.

Please also note the data extraction section in the methods and data extraction form (supplementary appendix 5) has been updated to include the extraction of information relating to anxiety:

Page 16-17:

2. Participant characteristics; method of depression assessment (e.g., structured clinical interview or screening tool), major depression diagnosis (yes/no), recruitment setting (clinical or community), age of mother (Mean, SD), age of infant (Mean, SD), time point of intervention (e.g., antepartum or postpartum), baseline anxiety, comorbidities, ethnicity of mother (n, %), relationship status (n, %), educational status (n, %), employment status (n, %), mothers first child (n, %), average household income (n, %), breastfeeding (n, %), level of depression severity at baseline (if clinical cut offs available), depression chronicity years (Mean, SD), number of participants invited, number of participants screened, number of participants eligible, number of participants randomised and reasons for non-eligibility.

34. For the qualitative extraction, I might consider listing the credentials of the two extractors as learning to code qualitative data and use NVivo requires some experience

The credentials of the researchers involved in the extraction have been added as follows:

Page 23:

Research supervisors (HOM, JW) have extensive experience in the analysis of qualitative data using a variety of approaches, are fully conversant in the use of NVivo, and will provide supervision to the PhD Student (DP) and Research Assistant (OB) conducting the meta-synthesis.

Dissemination Plan

35. The authors should consider providing specific examples of conferences/meetings/publications that you wish to disseminate your work at.

At this time in the process we wonder if it may be premature to name specific conferences and meetings in which we will disseminate. However, we have updated the dissemination plan to include the range of audiences that this review may be relevant to:

Page 24:

This review will be of relevance to a range of audiences including those working in the peripartum, women's health and CBT fields.

36. It may be helpful if rather than a single code for 'exclusion' you make the criteria more specific (e.g. 1) beyond 12 months postpartum, 2) no intervention, 3) not a CBT intervention etc.) That way when raters get together, they are easily able to identify why they (may) disagree

Thank you for this comment and highlighting the omission in the manuscript. During the full paper check process, a detailed form (see online supplementary appendix 2) is used, with raters reviewing each study against the PICOS statement and determining inclusion/exclusion for each item. When the raters meet to discuss discrepancies these forms are used to facilitate discussion concerning the items they disagree on. Overall reasons for exclusion will be recorded on the PRISMA flow chart. In addition, a more detailed exclusion table will be presented in the result manuscript with inclusion/exclusion presented for each specific criteria in line with the PICOS statement. Please see additions to the manuscript below:

Page 15-16:

Overall reasons for exclusion will be recorded (see online supplementary appendix 2) and reported on the PRISMA flow chart in the results manuscript. In addition, a more detailed exclusion table will be presented in the result manuscript with inclusion/exclusion presented for each specific criteria in line with the PICOS statement.

Summary

37. Additional rationale are required in the introductory section, and the authors do need to provide stronger arguments for why an update of Sockol's 2015 review is required at this point in time. The provision of more compelling arguments will highlight why this work is not simply just incremental. Moreover, it is important that the authors provide a more fulsome discussion of the secondary outcomes, their importance, and their amenability to treatment.

These points are addressed previous comments #4 and #9.

Searches

38. Searches are well done and take into account different regional spellings. Although, I recommend that they consider removal of the term 'alexithymia' as that is not synonymous with depression. The authors can also consider removing concept 4 (therapy terms) as this may unnecessarily restrict your search for all types of CBT interventions (and this is important to you as delivery type is a moderator you're interested in looking at). I can appreciate that this may reduce specificity (as you outlined on p.4 of the PRESS Guideline) although it may be better to be as inclusive as possible rather than exclusive if you want to ensure you capture all possible articles

Thank you for these comments. We would like to retain the term alexithymia as searches both with and without the term were trailed in pubmed and it added only one extra paper. Additionally, this term was specifically added by our peer reviewer during the peer reviewer processes. With regards to removing

concept 4 (therapy terms), this was something considered when developing the search syntax, however on balance the review team decided to keep the review more specific based on previous reviews in the area. We hope to guard against missing potentially eligible studies by searching a large number of electronic databases, contact with experts in the field, forward citation searches for all included studies using Google Scholar, hand searching references lists for all included studies, searching the five journals containing the highest numbers of included studies for recent potentially eligible publications in the past 12 months and searching grey literature databases.

Reviewer 2:

39. minor point - references 7 and 8 are the same This has been corrected; Page 25.